# Crowdsourcing to support training for public health: A scoping review

**Kadija M. Tahlil**[1], **Ucheoma Nwaozuru**[2], **Donaldson F. Conserve**[3], **Ujunwa F. Onyeama**[4], **Victor Ojo**[5], **Suzanne Day**[6], **Jason J. Ong**[7,8,9], **Weiming Tang**[10], **Nora E. Rosenberg**[4], **Titi Gbajabiamila**[5,11], **Susan Nkengasong**[9], **Chisom Obiezu-Umeh**[11], **David Oladele**[5,11], **Juliet Iwelunmor**[11], **Oliver Ezechi**[5], **Joseph D. Tucker**[6,10,12] *

1 Department of Epidemiology, University of North Carolina at Chapel Hill, Chapel Hill, North Carolina, United States of America, 2 Department of Implementation Science, Wake Forest University School of Medicine, Winston-Salem, North Carolina, United States of America, 3 Department of Prevention and Community Health, Milken Institute School of Public Health, The George Washington University, Washington, District of Columbia, United States of America, 4 Department of Health Behavior, University of North Carolina at Chapel Hill, Chapel Hill, North Carolina, United States of America, 5 Clinical Sciences Department, Nigerian Institute of Medical Research, Lagos, Nigeria, 6 Department of Medicine, Division of Infectious Diseases, University of North Carolina at Chapel Hill, Chapel Hill, North Carolina, United States of America, 7 Central Clinical School, Monash University, Melbourne, Australia, 8 Melbourne Sexual Health Centre, Alfred Health, Melbourne, Australia, 9 Department of Clinical Research, London School of Hygiene and Tropical Medicine, London, United Kingdom, 10 Institute of Global Health and Infectious Diseases, University of North Carolina at Chapel Hill, Chapel Hill, North Carolina, United States of America, 11 Department of Behavioral Science & Health Education, College for Public Health and Social Justice, Saint Louis University, St. Louis, Missouri, United States of America, 12 Clinical Research Department, Faculty of Infectious and Tropical Diseases, London School of Hygiene and Tropical Medicine, London, United Kingdom

* jdtucker@med.unc.edu

**Data Availability Statement:** All data can be found in the manuscript and supporting information files.

**Funding:** This work was supported by the Eunice Kennedy Shriver National Institute of Child Health

## Abstract

Crowdsourcing is an interactive process that has a group of individuals attempt to solve all or part of a problem and then share solutions with the public. Crowdsourcing is increasingly used to enhance training through developing learning materials and promoting mentorship. This scoping review aims to assess the literature on crowdsourcing for training in public health. We searched five medical and public health research databases using terms related to crowdsourcing and training. For this review, the concept of crowdsourcing included open calls, designathons, and other activities. We used a PRISMA checklist for scoping reviews. Each full-text was assessed by two independent reviewers. We identified 4,071 citations, and 74 studies were included in the scoping review. This included one study in a low-income country, 15 studies in middle-income countries, 35 studies in high-income countries, and 11 studies conducted in multiple countries of varying income levels (the country income level for 12 studies could not be ascertained). Nine studies used open calls, 35 used a hackathon, designathon or other "a-thon" event, and 30 used other crowdsourcing methods, such as citizen science programs and online creation platforms. In terms of crowdsourcing purpose, studies used crowdsourcing to educate participants (20 studies), develop learning materials (17 studies), enhance mentorship (13 studies) and identify trainees (9 studies). Fifteen studies used crowdsourcing for more than one training purpose. Thirty-four studies were done in-person, 31 were conducted virtually and nine used both meeting options for their crowdsourcing events. Seventeen studies generated open access materials. Our review found

and Human Development (NICHD) Grant number: UH3HD096929. KMT is supported by the National Institute of Allergy and Infectious Diseases (NIAID) Grant number: T32AI007001. JDT is supported by the National Institute of Allergy and Infectious Diseases (NIAID) Grant number: K24AI143471. The funders had no role in the design of the study; the collection, analysis, and interpretation of data; or the writing of the manuscript.

**Competing interests:** The authors have declared that no competing interests exist.

that crowdsourcing has been increasingly used to support public health training. This participatory approach can be a useful tool for training in a variety of settings and populations. Future research should investigate the impact of crowdsourcing on training outcomes.

## Introduction

In 2019, the World Health Organization (WHO) conducted a crowdsourcing open call to identify practical strategies to enhance research mentorship in low- and middle-income countries (LMICs) [1]. Open calls are an interactive form of crowdsourcing [2], which is a process that involves a group of individuals solving all or part of a problem, then sharing those solutions with the community [3]. The WHO open call solicited strategies to improve research mentorship and professional development, which were then assessed based on pre-specified criteria [1]. This open call received over 100 strategies, identified three individuals to contribute to a practical guide, engaged dozens of LMIC research institutions, and identified numerous open-access learning materials [1]. This underscores the ways that crowdsourcing approaches can enhance training and highlights the feasibility of crowdsourcing to enhance training engagement.

There is an increased recognition that we need to provide inclusive training to support diverse trainees [4]. There is a need to develop innovative approaches to identify early career investigators and nurture their opportunities for research [5], and to do so in participatory and inclusive ways [6]. Crowdsourcing approaches are one way to enhance training. Crowdsourcing has been previously used to identify LMIC researchers for training opportunities and engagement in health research as part of the WHO/TDR global programme [7, 8]. Crowdsourcing approaches have also been used in various other learning contexts, including developing learning materials [9], identifying open-access training resources [10], and identifying ways to enhance public health education and mentorship [1, 11, 12]. Training programs may benefit from crowdsourcing approaches that enhance community engagement, spur innovation, and identify learners. Traditional training programs often involve experts delivering one-way instruction and guidance to trainees with the aim of enhancing their personal and professional development. Crowdsourcing can be used to shift from conventional training approaches to a more open and collaborative process. Instead of experts being primarily responsible for training methods and outcomes, a diverse group of individuals from the community can work together to frame training strategies. Crowdsourcing to support training can help to prepare public health practitioners for interdisciplinary partnerships and provide access to community-developed resources.

Despite the growing interest in the potential of participatory approaches such as crowdsourcing in promoting training, few studies have examined the application of crowdsourcing for public health training purposes. There is little comprehensive understanding of the characteristic components of this approach for training, as well as its best practices, outputs, and outcomes. Although there are several empirical articles on the use of crowdsourcing approaches to promote public health training, no efforts have been made to collate and synthesize this body of knowledge. Amidst the growing importance of innovation in public health training, crowdsourcing approaches could potentially provide innovative and participatory training modalities and components. Hence, this scoping review investigates and summarizes the extent to which crowdsourcing has been used to support and promote public health training and explores critical components of how crowdsourcing can be used to improve public health training.

## Methods

### Search strategy

We organized a scoping review of the literature, drawing on the framework of Arksey and O'Malley [13] and following the PRISMA Extension for Scoping Reviews (S1 Checklist). We registered the protocol for the scoping review in the Open Science Framework (Registration DOI: https://doi.org/10.17605/OSF.IO/Q3PNH). A scoping approach was selected given substantial differences in the training methods and outcomes, several different ways of using crowdsourcing that preclude pooling, and many gaps in the literature. On March 14th, 2022, we conducted an initial search of five medical and public health research databases–PubMed, CINAHL, Embase, Global Health and Cochrane Library. We conducted a secondary search on April 5th, 2023 to capture any new articles published during the year after our initial search. The search algorithm included variations of the following terms: crowdsourcing, hackathon, designathon, training, education, mentorship, and capacity building (S1 Text). We identified and adapted these search terms from prior crowdsourcing and training review literature. Included publications focused on using crowdsourcing methods and training. We used the WHO/TDR definition of crowdsourcing: "the process of having a large group, including experts and non-experts, solve a problem and then share the solution with the public [14]." This definition is grounded in a broader crowdsourcing set of approaches [15]. This includes open calls (also known as innovation challenges or contests), designathons (also known as hackathons or sprint-like events), and other forms of crowdsourcing. For this paper, we define training broadly to encompass formal education, informal education, mentorship, coaching, and capacity-building for a wide range of ages and learning backgrounds [16]. We exported records from our search, removed duplicates using EndNote X9, and performed online screening.

### Study selection

Inclusion criteria were the following: relevant to public health; clear description of crowdsourcing methodology; the overall purpose was to enhance training, education, mentorship, or a related area. This included empirical descriptions of crowdsourcing training programs, clinical trials that used crowdsourcing methods for education or training, and descriptions of methods. There were no geographic or time restrictions on the search. Studies, commentaries/editorials, and opinion pieces that described potential programs that have not been implemented were excluded. We excluded systematic, scoping, or narrative reviews and studies that were not written in English.

Two reviewers independently reviewed titles and abstracts for inclusion, and a third reviewer was available to resolve discrepancies. Then two independent reviewers examined full-text manuscripts, excluding studies based on the criteria above. Data were extracted about the purpose of crowdsourcing and health topic of the crowdsourcing event. We also extracted information on the following components of crowdsourcing that can support public health training: crowdsourcing method (open call, designathon, other), country income level, type of crowdsourcing event (digital, in-person or both), and whether open access materials were generated. We categorized crowdsourcing based on the Joint International Consensus Statement on Crowdsourcing [17]. Subsequently, a narrative synthesis of the extracted data was performed. The stages of our narrative synthesis included: 1) descriptive statistics to summarize the extent and nature of included studies and 2) thematic categorization, which involved identifying common training areas and grouping studies based on those training categories.

## Results

### Identification of studies

We retrieved 3,438 publications from our initial database searches. After removing duplicates, the publications were screened for the relevance of the title and abstract, resulting in the exclusion of 2,108 publications. We further evaluated 88 publications for full-text eligibility. Of these, 28 articles were excluded for the following reasons: insufficient description of crowdsourcing methods (n = 8), not focused on crowdsourcing (n = 6), insufficient details on training, mentorship or education (n = 6), wrong article type (n = 3), not written in English (n = 1), and same crowdsourcing event already described in another study (n = 4). We retrieved 633 publications from our secondary database searches. The total number of publications retrieved from the initial and secondary database searches was 4,071. Overall, 74 studies were included in this review (Fig 1).

### Descriptive characteristics

Table 1 shows the descriptive characteristics of the 74 studies included in this review. The majority of studies were conducted in high-income countries (56.5%), followed by middle-

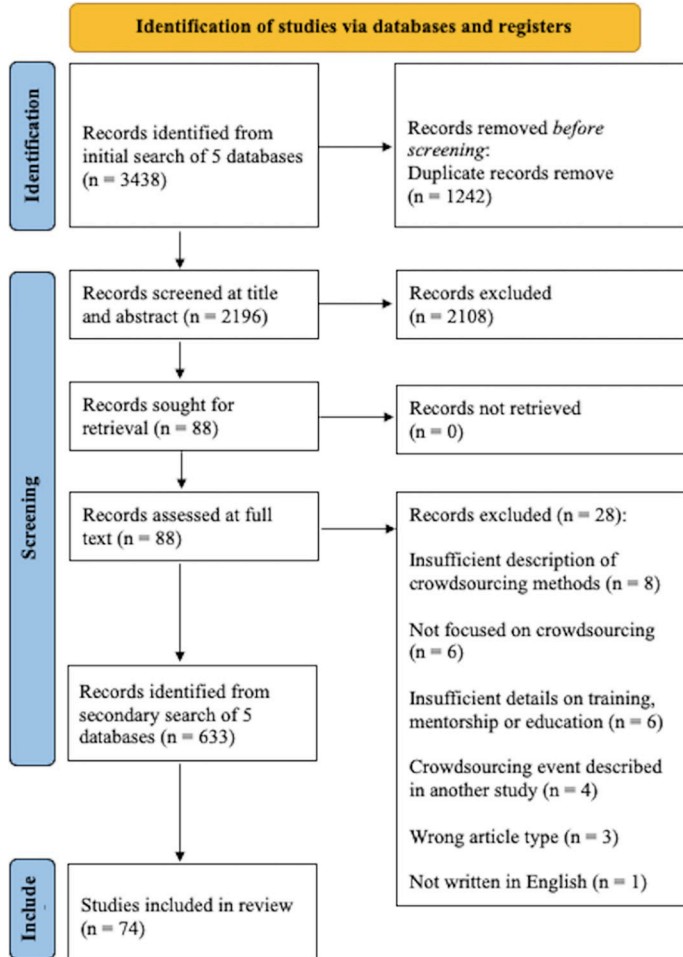

**Fig 1. PRISMA flow diagram.** The total number of records identified from the initial (n = 3,438) and secondary (n = 633) searches is 4,071. The secondary search was performed after records from the initial search were assessed at full text review.

**Table 1. Characteristics of studies using crowdsourcing for public health training between 2011–2023 (N = 74).**

| | n | % |
|---|---|---|
| **Country Income Classification** | | |
| Low-income | 1 | 1.6 |
| Middle-income | 15 | 24.2 |
| High-income | 35 | 56.5 |
| Multiple income levels (Multiple countries) | 11 | 17.7 |
| Missing | 12 | |
| **Crowdsourcing Method** | | |
| Hackathon/Designathon/Other "a-thon" events | 35 | 47.3 |
| Open call | 9 | 12.2 |
| Other | 30 | 40.5 |
| **Type of Crowdsourcing Event** | | |
| In-person | 34 | 45.9 |
| Virtual | 31 | 41.9 |
| Both | 9 | 12.2 |
| **Purpose of Crowdsourcing** | | |
| Education | 20 | 27.0 |
| Develop learning materials | 17 | 23.0 |
| Promote mentorship | 13 | 17.6 |
| Identify trainees | 9 | 12.2 |
| Multi-purpose | 15 | 20.3 |
| **Open Access Materials Generated** | | |
| No | 57 | 77.0 |
| Yes | 17 | 23.0 |

income countries (24.2%). Eleven studies (17.7%) used crowdsourcing approaches in multiple countries of varying income levels and one study (1.6%) was conducted in a low-income country. Thirty-five studies (47.3%) used a hackathon or other "a-thon" event as the crowdsourcing method. Thirty studies (40.5%) used crowdsourcing methods other than open calls, hackathons, or other "a-thon" events. These other methods mainly included online creation platforms, citizen science programs, and peer groups. The majority of these crowdsourcing events took place in-person (45.9%). Only 17 studies (23%) identified in this review generated open access materials. The majority of studies (81%) were published in 2018 or after.

## Training areas

We identified the use of crowdsourcing to support four areas of public health training (Fig 2). Crowdsourcing was used to 1) educate participants, 2) develop learning materials, 3) promote mentorship, or 4) identify trainees. Below, we describe the role of crowdsourcing for each of these training areas in further detail.

## Crowdsourcing to educate participants

Twenty studies (27%) used crowdsourcing to enhance participants' education and capacity-building (Table 2) [18–37]. Of these studies, 12 were solely conducted in middle- or high-income countries. Twelve studies educated participants in-person, six were done virtually, and two used both methods. Eight studies used a hackathon or datathon to educate participants and five studies utilized citizen science programs. Hackathons were used to deliver a variety of educational content, including healthcare innovation [18], social work education [36], and

# CROWDSOURCING FOR TRAINING

## Did you know?

**Crowdsourcing has been used to identify learners and trainees. It has also been adapted to provide participants with opportunities to learn essential skills and launch their ideas from mentors. Crowdsourcing can also be used to educate participants and develop learning materials**

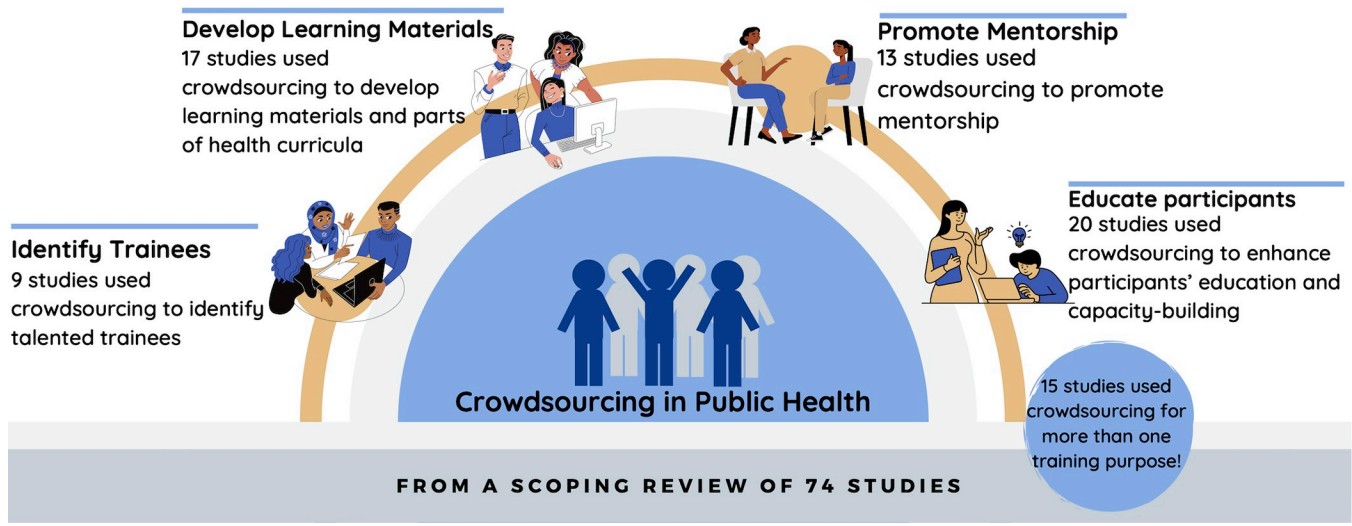

**Develop Learning Materials**
17 studies used crowdsourcing to develop learning materials and parts of health curricula

**Promote Mentorship**
13 studies used crowdsourcing to promote mentorship

**Identify Trainees**
9 studies used crowdsourcing to identify talented trainees

**Educate participants**
20 studies used crowdsourcing to enhance participants' education and capacity-building

**Crowdsourcing in Public Health**

15 studies used crowdsourcing for more than one training purpose!

**FROM A SCOPING REVIEW OF 74 STUDIES**

**Fig 2. Crowdsourcing for public health training infographic.**

neuroscience [20]. The Mount Sinai Health Hackathon, which is a 48-hour team-based competition that occurs annually, was a crowdsourcing event that serves as a model for team science education [23, 26]. This hackathon brought together individuals from different disciplines to work on a shared health problem, which fostered an environment of experiential learning through collaboration and communication. Also, three of the five studies that used citizen science programs to crowdsource for education were focused on environmental health. These studies educated high school students on topics such as air pollution [21], radiation monitoring [24], and radon exposure [28]. Upon completion of the educational portion of the programs, these students engaged in citizen science by going into their communities and collecting and reporting environmental data.

## Crowdsourcing to develop learning materials

Seventeen studies (23%) used crowdsourcing to develop learning materials and parts of health curricula (Table 3) [11, 38–53]. Of these studies, 11 were solely conducted in high-income countries. Fourteen studies only hosted the crowdsourcing event virtually. Ten studies made their learning materials publicly available. Nine studies utilized online creation platforms where participants collaborated to produce, review, collate and share learning materials. The use of online creation platforms resulted in the development of various learning resources, such as demonstration videos for learners on performing common pediatric procedures in

**Table 2. Summary of studies that used crowdsourcing to educate participants (N = 20).**

| First Author | Publication Year | Country Income Level* | Number of Participants | Crowdsourcing Method | Type of Event | Open Access Materials Generated | Health Topic |
|---|---|---|---|---|---|---|---|
| Blindenbach-Driessen, F. | 2014 | Not specified | 30 | Hackathon | In-person | No | Healthcare innovation |
| Cai, H. | 2022 | MIC | 172 | Classroom curriculum | In-person | No | Global health |
| Craddock, RC. | 2016 | MIC/HIC | Not specified | Hackathon | In-person | Yes | Neuroscience |
| Ellenbug, JA. | 2019 | LIC/MIC/HIC | ~11200 | Citizen science program | Both | Yes | Air pollution |
| Estes, CF. | 2021 | Not specified | 3817 | Online creation platform | Virtual | Yes | Radiation oncology |
| Fattah, L. | 2020 | HIC | 76 | Hackathon | In-person | No | Rare diseases |
| Fojtíková, I. | 2019 | HIC | Over 30 | Citizen science program | In-person | No | Radiation monitoring |
| Fuhrmeister, ER. | 2021 | HIC | Not specified | Classroom project | In-person | No | Antimicrobial resistance |
| Gabrilove, JL. | 2018 | HIC | 87 | Hackathon | In-person | No | Cancer |
| González, SA. | 2022 | MIC | 97 | Citizen science program | In-person | No | Health promotion |
| Hahn, EJ. | 2020 | HIC | 27 | Citizen science program | In-person | No | Radon exposure |
| Ischia, J. | 2019 | Not specified | Not specified | Online creation platform | Virtual | No | Cancer |
| Matthews, AK. | 2022 | HIC | 8 | Citizen science program | Virtual | No | Cancer |
| Piza, FMT. | 2018 | MIC | 49 | Datathon | In-person | No | Healthcare databases |
| Preiksaitis, C. | 2023 | HIC | 12 | Hackathon | In-person | No | Medical innovation |
| Puius, YA. | 2023 | HIC | Not specified | Work group | Virtual | Yes | Infectious diseases |
| See, C. | 2014 | Not specified | Not specified | Online and offline creation platforms | Both | No | Medical education |
| Sherbino, J. | 2015 | Not specified | 86 | Online journal club | Virtual | No | Medical education |
| Wilson, J. | 2019 | HIC | 32 | Hackathon | In-person | No | Homelessness |
| Zou, Y. | 2022 | LIC/MIC/HIC | Not specified | Hackathon | Virtual | No | Radiation oncology |

*LIC = low-income country; MIC = middle-income country; HIC = high-income country

resource-constrained settings [40], a cancer genetics e-textbook co-created by and for undergraduate students [49], a healthcare curriculum as part of a diagnostic radiography programme [50], and a mobile vascular surgery handbook that can help users access information during conferences and clinical care [52]. Also, one study that used crowdsourcing to develop learning materials was a global open call to solicit questions, infographics, and images to create open-access materials on antimicrobial resistance [11]. High-scoring entries from this open call were shared as learning resources with the public.

**Crowdsourcing to promote mentorship.** Thirteen studies (17.6%) used crowdsourcing to promote mentorship (Table 4) [1, 8, 9, 13, 54–62]. Five of these studies were conducted in high income countries. Eight studies were conducted virtually, four were done in-person, and one used both methods to host their event. Eight studies used hackathons, three used open calls, and two used peer groups as their crowdsourcing method. All eight studies that organized hackathons provided mentors for their participants [8, 9, 54, 55, 58, 60–62]. The mentors had varying roles, which included providing expert knowledge, training participants in public speaking and presentation skills, reviewing ideas and prototypes, offering encouragement and support, and connecting participants to their networks. Two studies initiated open calls related to research mentorships; one solicited ideas to enhance research mentorship in LMICs [1] and another gathered data to understand and improve the impact of a global health research training program on trainees and students [13]. Peer groups were also vehicles to promote

**Table 3. Summary of studies that used crowdsourcing to develop learning materials (N = 17).**

| First Author | Publication Year | Country Income Level* | Number of Participants | Crowdsourcing Method | Type of Event | Open Access Materials Generated | Health Topic |
|---|---|---|---|---|---|---|---|
| Adsul, P. | 2023 | HIC | Not specified | Work group | Virtual | Yes | Health equity |
| Bate, A. | 2017 | MIC/HIC | Not specified | Online creation platform | Virtual | No | Pharmacovigilance education |
| Bensman, RS. | 2017 | HIC | Not specified | Online creation platform | Virtual | Yes | Global health |
| Berk, J. | 2018 | HIC | Not specified | Online creation platform | Virtual | Yes | Medical education |
| Drasovean, Y. | 2021 | Not specified | Over 2500 | Open call | Virtual | No | Coronavirus disease 2019 |
| Herodotou, C. | 2018 | HIC | Not specified | Online creation platform | Both | Yes | Citizen inquiry |
| Ianni, PA. | 2020 | HIC | Not specified | Online creation platform | Virtual | No | Translational science |
| Ilagan-Ying, YC. | 2022 | HIC | 67 | Clinic education development | In-person | No | Medical education |
| Kercheval, JB. | 2021 | HIC | 42 | Curriculum development | Virtual | No | Medical education |
| Kpokiri, EE. | 2021 | MIC/HIC | Not specified | Open call | Virtual | Yes | Antimicrobial resistance |
| Leonard, HL. | 2023 | LIC/MIC/HIC | 49 | Hackathon | Virtual | No | Parkinson's disease |
| Seam, N. | 2019 | HIC | Not specified | Online creation platform | Virtual | Yes | Medical education |
| Seid-Karbasi, P. | 2017 | HIC | 89 | Online creation platform | Both | Yes | Cancer genetics |
| St John-Matthews, J. | 2020 | HIC | 27 | Online creation platform | Virtual | No | Radiography |
| Staziaki, PV. | 2022 | Not specified | Not specified | Hackathon | Virtual | Yes | Radiology education |
| Sutzko, DC. | 2019 | HIC | 54 | Online creation platform | Virtual | Yes | Vascular surgery |
| Tangcharoensathien, V. | 2020 | Not specified | 1483 | Global online consultation | Virtual | Yes | Infodemics |

*LIC = low-income country; MIC = middle-income country; HIC = high-income country

mentorship for internal medicine residents [56, 57]. Peer mentorships were used to promote professional development and support resident well-being.

## Crowdsourcing to identify trainees

Nine studies (12.2%) used crowdsourcing methods to identify talented trainees for further opportunities, making training programs more inclusive (Table 5) [63–71]. Of these nine studies, three used open calls, four used hackathons, and two used citizen science programs to identify trainees. Five studies were conducted in-person, one was done virtually, and three used both methods to implement the crowdsourcing event. Five studies were solely conducted in high-income countries. Previous open calls that used crowdsourcing to identify trainees invited key stakeholders to offer their solutions on how to increase women's participation in infectious diseases research fellowships [69], how to promote HIV self-testing among young people in Nigeria [70], how to improve the use of evidence-based practices in a behavioral health system [71]. In these past crowdsourcing open calls, participants with high-quality and promising ideas were invited to refine, finalize, present, and potentially implement their

**Table 4. Summary of studies that used crowdsourcing to promote mentorship (N = 13).**

| First Author | Publication Year | Country Income Level* | Number of Participants | Crowdsourcing Method | Type of Event | Open Access Materials Generated | Health Topic |
|---|---|---|---|---|---|---|---|
| Bao, H. | 2020 | LIC/MIC/HIC | 44 | Open call | Virtual | No | Research mentorship |
| Bolton, WS. | 2021 | HIC | 123 | Hackathon | Virtual | Yes | Coronavirus disease 2019 |
| Braune, K. | 2021 | HIC | 48 | Hackathon | Virtual | No | Coronavirus disease 2019 |
| Buteau, A | 2019 | Not specified | Not specified | Peer group | In-person | No | Professional development |
| Ciccariello, C. | 2018 | Not specified | 180 | Peer group | Both | No | Well-being in residency |
| DePasse, JW. | 2014 | HIC | Not specified | Hackathon | In-person | No | Healthcare innovation |
| Goel, S. | 2022 | MIC | 150+ | Open call | Virtual | No | Hypertension |
| Koszalinski, RS. | 2021 | Not specified | 1812 | Hackathon | Virtual | No | Coronavirus disease 2019 |
| Oppong, E. | 2021 | LIC/MIC | 123 | Open call | Virtual | No | Research mentorship |
| Poncette, AS. | 2020 | HIC | 30 | Hackathon | In-person | No | Healthcare innovation |
| Ruzgar, NM. | 2020 | HIC | 31 | Hackathon | In-person | No | Surgery |
| Tan, RKJ. | 2022 | MIC | Not specified | Designathon | Virtual | No | Global health |
| Ulitin, A. | 2022 | MIC | 28 | Hackathon | Virtual | No | Coronavirus disease 2019 |

*LIC = low-income country; MIC = middle-income country; HIC = high-income country

solutions through feedback and collaboration with content experts. These crowdsourcing methods were inclusive as they aggregated the ideas and perspectives of the trainees with the knowledge and experiences of the experts.

**Table 5. Summary of studies that used crowdsourcing to identify trainees (N = 9).**

| First Author | Publication Year | Country Income Level* | Number of Participants | Crowdsourcing Method | Type of Event | Open Access Materials Generated | Health Topic |
|---|---|---|---|---|---|---|---|
| Amat, M. | 2021 | HIC | 98 | Hackathon | In-person | No | Healthcare challenges |
| Askins, N. | 2020 | HIC | Not specified | Citizen science program | Virtual | No | Cancer |
| Cooper, K. | 2018 | HIC | 200 | Hackathon | In-person | No | Radiology |
| Fadlelmola, FM. | 2021 | MIC | 24 | Hackathon | In-person | No | Genomic medicine and microbiome |
| Hidalgo-Ruz, V. | 2013 | HIC | 983 | Citizen science program | In-person | No | Marine environment research |
| Li, C. | 2020 | MIC | 38 | Hackathon | In-person | No | Health care utilization |
| Liu, E. | 2020 | LIC/MIC/HIC | Not specified | Open call | Both | No | Infectious diseases research |
| Rosenberg, NE. | 2021 | MIC | 769 | Open call | Both | No | Human immunodeficiency virus |
| Stewart, RE. | 2019 | HIC | 55 | Open call | Both | No | Evidence-based clinical practices |

*LIC = low-income country; MIC = middle-income country; HIC = high-income country

## Crowdsourcing for more than one training purpose

Fifteen studies (20.3%) used crowdsourcing for more than one training purpose (Table 6) [72–86]. Of these studies, thirteen used hackathons or other "a-thon" events to achieve their training goals. Twelve studies were conducted in either a middle- or high-income country. All but three study's crowdsourcing events were hosted in-person. The multiple training purposes in

**Table 6. Summary of studies that used crowdsourcing for more than one training purpose (N = 15).**

| First Author | Publication Year | Country Income Level* | Number of Participants | Training Purpose | Crowdsourcing Method | Type of Event | Open Access Materials Generated | Health Topic |
|---|---|---|---|---|---|---|---|---|
| Babatunde, A. | 2023 | LIC/MIC | 44 | Identify trainees; Promote mentorship | Hackathon | Virtual | No | Health equity |
| Butt, WA. | 2020 | LIC | 116 | Identify trainees; Promote mentorship | Hackathon | In-person | No | Medical education |
| Callisto, D. | 2018 | MIC | Not specified | Learning materials; Promote mentorship | Hackathon | In-person | No | Human immunodeficiency virus; viral hepatitis |
| Euzébio De Lima, C. | 2016 | MIC | Not specified | Education; Promote mentorship | Hackathon | In-person | No | Human immunodeficiency virus; Sexually transmitted diseases |
| Hawk, M. | 2017 | MIC | 24 | Identify trainees; Promote mentorship | Grantathon | In-person | No | Mental health |
| Jordan, RC. | 2011 | HIC | 82 | Identify trainees; Education | Citizen science program | In-person | No | Conservation science |
| Kahn, MJ. | 2022 | Not specified | 37 | Education; Promote mentorship | Open call | Virtual | No | Medical education |
| Lewis, S. | 2021 | MIC | 90 | Education; Promote mentorship | Hackathon | In-person | No | Cancer |
| Pathanasethpong, A. | 2017 | MIC | 140 | Education; Promote mentorship | Hackathon | In-person | Yes | Mobile health technology |
| Ramadi, KB. | 2019 | HIC | 72 | Identify trainees; Promote mentorship | Hackathon | In-person | No | Health diplomacy |
| Schwerdtle, P. | 2018 | HIC | 300 | Education; Learning materials | Mapathon | Both | Yes | Global health |
| Silver, JK. | 2016 | HIC | 102 | Education; Promote mentorship | Hackathon | In-person | No | Rehabilitation medicine innovation |
| Tahlil, KM. | 2021 | MIC | 42 | Identify Trainees; Promote mentorship | Designathon | In-person | No | Human immunodeficiency virus |
| Wang, JK. | 2018 | HIC | 257 | Education; Promote mentorship | Hackathon | In-person | No | Clinical needs |
| Wang, JK. | 2018 | MIC/HIC | 245 | Education; Promote mentorship | Hackathon | In-person | No | Medical innovation |

*LIC = low-income country; MIC = middle-income country; HIC = high-income country

13 of these studies included promoting research mentorship. One study used a crowdsourcing workshop in India to identify trainees and build their capacity to provide mental health services [76]. These trainees were identified after applying for the workshop and then paired with mentor experts who guided them through developing a mental health research funding proposal.

## Discussion

This scoping review describes the extent and characterizes existing research on crowdsourcing for public health training. We found that crowdsourcing has been used to support four areas of training: to educate participants, develop learning materials, promote mentorship, and identify trainees. Studies in this review featured different crowdsourcing approaches to improve public health training. We found that these participatory approaches have supported training on a broad range of health topics, including environmental health [21, 24, 28], infectious diseases [11, 25, 54, 55, 60, 69, 70, 74, 75, 84], and mental health [76]. This scoping review extends the literature on crowdsourcing to examine how it has been used to benefit public health training.

The findings of this scoping review provide an evidence base for the role of crowdsourcing to foster education and develop learning materials. These studies used a variety of crowdsourcing approaches such as hackathons, citizen science programs at schools, and online platforms to deliver education. These studies suggest that crowdsourcing can provide a structured method for community health and medical education. We also found that all studies that used crowdsourcing to develop learning materials used online collaboration systems. This demonstrates that crowdsourcing can be used to engage diverse online communities and provide a virtual environment where these communities can work together to develop training resources. Moreover, online crowdsourcing approaches may be less resource-intensive than in-person crowdsourcing events, which could be potentially useful in resource-limited settings. Crowdsourcing comes with an obligation to give back to the public who created the idea. Therefore, learning materials that were developed from crowdsourcing can then be used for future education and capacity-building. In the 13 studies that used crowdsourcing to develop learning materials, eight studies made their materials widely available to the public [11, 40, 41, 43, 48, 49, 52, 53], which allows the resources to be accessible without restrictions.

We also found that crowdsourcing has been used to promote research and professional mentorship. A recurring theme among studies focused on promoting mentorship is the use of participatory approaches to increase community participation in research and development of interventions. Crowdsourcing events have been used to identify participants with promising ideas and connect them with mentors who can help them iteratively refine their ideas. Typically, this would include events where participants practice pitching their ideas to these mentors and receive tailored feedback. Also, an additional advantage that can arise from these crowdsourcing events is the opportunity for mentees to have access to their mentors' networks. This can help mentees strengthen their research ideas and broaden their own professional networks. It is also important for public health trainers to carefully consider and employ the most appropriate crowdsourcing method to enhance mentorship. Hackathons, for example, are a popular crowdsourcing approach to provide mentorship. Considerations for hackathons to successfully promote mentorship can include determining the appropriate mentor-mentee ratio for the event and incorporating designated and adequate mentoring sessions into the hackathon agenda.

Moreover, we found that crowdsourcing is a useful approach to identifying and engaging talented trainees. Open calls, in particular, can be used to select individuals for training. For

example, one study used a crowdsourcing open call to solicit ideas on how to improve HIV self-testing among youth [70]. Participants with promising ideas were selected as finalists and proceeded to subsequent stages of the study for further development of their ideas and opportunities for apprenticeships. Another study used an open call to solicit ideas on improving the participation of women in an infectious disease research fellowship [69]. The top ideas from this open call were then implemented in the fellowship's next application cycle, which saw an increase in the number of women applicants. Open calls have the potential to reach a wide and varied audience, which can provide an opportunity for enhanced inclusivity and engagement of early investigators. These methods should be considered by public health researchers as suitable approaches to identify and engage with early investigators.

Our scoping review had several limitations. First, the crowdsourcing literature is diverse in form and content, making the pooling of studies for meta-analysis difficult. Second, we captured fewer studies from low- and middle-income countries. This may be the result of fewer crowdsourcing activities in those countries or less reporting of those experiences. Third, our scoping review did not include grey literature, thus relevant studies may have been missed. Fourth, we may not have captured some capacity-building programs that used crowdsourcing but were not formally evaluated.

Our findings have implications for public health research, programming, and education. Crowdsourcing can serve as an innovative model to advance public health training. Crowdsourcing provides a way to go beyond conventional didactic approaches to training to engaging and collaborative methods to training. This may result in public health professionals that are prepared to develop creative and novel solutions to address challenging public health issues. These participatory approaches can be considered for use by public health agencies looking to identify and provide funding opportunities for talented early investigators, educational institutions that are preparing public health students for the workforce, or organizations that seek to provide opportunities for professional development and mentorship.

## Conclusion

Our scoping review found a wide range of studies supporting the use of crowdsourcing methods for training in public health. Future research should evaluate the impact of crowdsourcing on training outcomes.

## Supporting information

**S1 Checklist. Preferred Reporting Items for Systematic reviews and Meta-Analyses extension for Scoping Reviews (PRISMA-ScR) checklist.**
(DOCX)

**S1 Text. Search algorithm on crowdsourcing to support public health training.**
(DOCX)

## Author Contributions

**Conceptualization:** Joseph D. Tucker.

**Formal analysis:** Kadija M. Tahlil, Ucheoma Nwaozuru, Donaldson F. Conserve, Ujunwa F. Onyeama.

**Investigation:** Kadija M. Tahlil, Ucheoma Nwaozuru, Donaldson F. Conserve, Victor Ojo.

**Project administration:** Joseph D. Tucker.

**Visualization:** Ujunwa F. Onyeama.

**Writing – original draft:** Kadija M. Tahlil, Ucheoma Nwaozuru, Donaldson F. Conserve, Ujunwa F. Onyeama, Joseph D. Tucker.

**Writing – review & editing:** Kadija M. Tahlil, Ucheoma Nwaozuru, Donaldson F. Conserve, Ujunwa F. Onyeama, Victor Ojo, Suzanne Day, Jason J. Ong, Weiming Tang, Nora E. Rosenberg, Titi Gbajabiamila, Susan Nkengasong, Chisom Obiezu-Umeh, David Oladele, Juliet Iwelunmor, Oliver Ezechi, Joseph D. Tucker.

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
