## [Decision Letter · Decision Letter 0]

20 Mar 2023

PGPH-D-22-01976

Crowdsourcing to support training for public health: A scoping review

Dear Dr. Tucker,

Thank you for submitting your manuscript to PLOS Global Public Health. We thank you for your patience regarding the length of time it's taken to get back to you with reviewer comments. After careful consideration, we feel that it has merit but does not fully meet PLOS Global Public Health’s publication criteria as it currently stands. Therefore, we invite you to submit a revised version of the manuscript that addresses the points raised during the review process.

You'll see both sets of reviewers' comments below. I would ask you to pay particularly close attention in replying to Reviewer One's comments and questions given their questioning of the paper's methods and methodology.

We look forward to receiving your revised manuscript.

Kind regards,

Diego S Silva, PhD

Academic Editor

Journal Requirements:

1. In the online submission form, you indicated that "The data used and analyzed during this study are available from the corresponding author upon reasonable request.". All PLOS journals now require all data underlying the findings described in their manuscript to be freely available to other researchers, either 1. In a public repository, 2. Within the manuscript itself, or 3. Uploaded as supplementary information.

2. Please add a full list of legends for all your Supporting Information files after the references list.

3. Some material included in your submission may be copyrighted. According to PLOS’s copyright policy, authors who use figures or other material (e.g., graphics, clipart, maps) from another author or copyright holder must demonstrate or obtain permission to publish this material under the Creative Commons Attribution 4.0 International (CC BY 4.0) License used by PLOS journals. Please closely review the details of PLOS’s copyright requirements here: PLOS Licenses and Copyright. If you need to request permissions from a copyright holder, you may use PLOS's Copyright Content Permission form.

Potential Copyright Issues:

Fig 2: Please confirm whether you drew the images / clip-art within the figure panels by hand. If you did not draw the images, please provide (a) a link to the source of the images or icons and their license / terms of use; or (b) written permission from the copyright holder to publish the images or icons under our CC-BY 4.0 license. Alternatively, you may replace the images with open source alternatives. See these open source resources you may use to replace images / clip-art:

- https://openclipart.org/

Additional Editor Comments (if provided):

Reviewers' comments:

Reviewer's Responses to Questions

**Comments to the Author**

1. Does this manuscript meet PLOS Global Public Health’s publication criteria? Is the manuscript technically sound, and do the data support the conclusions? The manuscript must describe methodologically and ethically rigorous research with conclusions that are appropriately drawn based on the data presented.

Reviewer #1: Partly

Reviewer #2: Yes

2. Has the statistical analysis been performed appropriately and rigorously?

Reviewer #1: N/A

Reviewer #2: Yes

3. Have the authors made all data underlying the findings in their manuscript fully available (please refer to the Data Availability Statement at the start of the manuscript PDF file)?

Reviewer #1: Yes

Reviewer #2: Yes

4. Is the manuscript presented in an intelligible fashion and written in standard English?

Reviewer #1: Yes

Reviewer #2: Yes

5. Review Comments to the Author

Reviewer #1: The authors have made a substantial effort to map the evidence on the subject, which deserves recognition. However, there are methodological issues. Additionally, the main message is not clear. Main points:

1. The manuscript lacks a registered study protocol as recommended by the latest scoping review guidelines (Tricco et al., 2018).

2. Could the authors make the introduction punchier? Why is the crowdsourcing approach crucial? Is it more economical? Does it have a greater engagement capacity than other approaches? Does it have greater capillarity? Why is this an important topic for the field of global health? Would it support especially low- and middle-income countries? Would this type of approach help with fast responses to public health emergencies? In which way?

3.The research question is not clearly described.

4. There are updated guidelines on how to conduct scoping reviews. Please see in: doi: 10.11124/JBIES-21-00064; doi: 10.1186/s13012-022-01223-6; doi: 10.7326/M18-0850. The authors chose a framework from 2003 [Arksey and O’Malley, 2003]. The authors should justify why they chose this approach instead of the more updated ones or describe their efforts to guarantee the manuscript is adjusted to reflect the latest and updated recommendations.

5. The authors should clarify how search terms were identified. Are they all "medical subject headings," merely keywords, or both?

6. The authors should describe narrative synthesis in detail, mainly: (1) Which information was extracted from the publications? What methodology do authors apply to gather results? (2) What categories/domains emerged from the reading? (3) Who extracted the content (only one author)? Two authors? (4) If disagreements surge about including or not a text fragment, how did the author solve it?

7.The discussion section did not adequately contextualize the findings and discuss related research.

Reviewer #2: Well done to the authors for conducting this review this interesting review manuscript on crowdsourcing for public health. This is a critical and timely piece that presents valuable evidence for employing these strategies especially in public health training. Please see below for my comments to the authors

Abstract:

Line 52: search terms related to crowdsourcing and training alone? Public health?

Line 52: Search done on March 14th, 2022, given it’s over a year now, consider repeating search to pick up any new articles relevant to the review

Line 53: reword the sentence to read ---For this review, the concept of crowdsourcing included open calls, designathons, and other activities. For the others is there a definite list of what other activities were considered as crowdsourcing for this SR?

Introduction:

Lines 71-82: the opening paragraph relies heavily on the WHO TDR open call on mentorship. This is good and sets the scene but would be great to also emphasize crowdsourcing as being an innovative approach to support training and some of the benefits this offers.

Line 85: typo here at the beginning of the paragraph- change “the” to “there”

Line 88-89: crowdsourcing has been used to identify LMIC researchers for training in crowdfunding and public engagement in health research as part of a WHO/TDR global programme

Methods:

Line 113: consider updating the search to pick up any new articles relevant to the review

Line 141-142: Please add a sentence detailing how the extracted was analysed into review findings and presented (add references as relevant)

Was the review registered as a protocol in an open access repository?

Results:

Table 1: first- The section on purpose says 13 studies developed learning materials and the next session says 14 studies generated open access materials? Secondly, I am not sure of the variable “open access materials generated” in table 1 since not all papers included had the purpose of generating materials.

Discussion:

Line 303: 13 studies used crowdsourcing to develop learning materials, eight developed open access materials, this might be confusing for the reader going with the results presented in table 1 that suggests 14 included articles developed open access materials. Please review the presentation of these results.

References:

Line 380-383: Refs 8 and 9 appear to be incomplete with title of the articles missing

6. PLOS authors have the option to publish the peer review history of their article (what does this mean?). If published, this will include your full peer review and any attached files.

**Do you want your identity to be public for this peer review?** For information about this choice, including consent withdrawal, please see our Privacy Policy.

Reviewer #1: No

Reviewer #2: **Yes: **Eneyi E. Kpokiri

---

## [Decision Letter · Decision Letter 1]

1 Jun 2023

PGPH-D-22-01976R1

Crowdsourcing to support training for public health: A scoping review

Dear Dr. Tucker,

Thank you for submitting your manuscript to PLOS Global Public Health. After careful consideration, we feel that it has merit but does not fully meet PLOS Global Public Health’s publication criteria as it currently stands. Therefore, we invite you to submit a revised version of the manuscript that addresses the points raised during the review process.

At this point, it's just some minor but important points we'd like you to address from Reviewer 1. We appreciate your patience with the time it's taken to review your paper and we look forward to accepting it pending these final revisions.

We look forward to receiving your revised manuscript.

Kind regards,

Diego S Silva, PhD

Academic Editor

Journal Requirements:

Additional Editor Comments (if provided):

Reviewers' comments:

Reviewer's Responses to Questions

**Comments to the Author**

1. If the authors have adequately addressed your comments raised in a previous round of review and you feel that this manuscript is now acceptable for publication, you may indicate that here to bypass the “Comments to the Author” section, enter your conflict of interest statement in the “Confidential to Editor” section, and submit your "Accept" recommendation.

Reviewer #1: All comments have been addressed

2. Does this manuscript meet PLOS Global Public Health’s publication criteria? Is the manuscript technically sound, and do the data support the conclusions? The manuscript must describe methodologically and ethically rigorous research with conclusions that are appropriately drawn based on the data presented.

Reviewer #1: Partly

3. Has the statistical analysis been performed appropriately and rigorously?

Reviewer #1: N/A

4. Have the authors made all data underlying the findings in their manuscript fully available (please refer to the Data Availability Statement at the start of the manuscript PDF file)?

Reviewer #1: Yes

5. Is the manuscript presented in an intelligible fashion and written in standard English?

Reviewer #1: Yes

6. Review Comments to the Author

Reviewer #1: The authors have diligently addressed the raised comments and updated the revised guidelines and search strategy. These have significantly improved the paper's methodology. Nonetheless, additional revisions are still necessary. The implementation of the following recommendations would be greatly appreciated.

1. Introduction (lines 80-82): The definition of training should be moved from the introduction to the methods section.

2. Methods section:

2.1. Please, review the PRISMA-ScR Checklist. The authors answered 'no' to the following statement: "Indicate whether a review protocol exists; state whether and where it can be accessed (e.g., via a Web address); and, if available, provide registration information, including the registration number."

2.2. S2 File and PRISMA FLOWCHART. Please ensure the reader can follow the final results considered. The results available in S2 File need to match the results in the box 'Records identified from the initial search´. I am wondering if the authors considered the results #3; #1, and #2 for each database. If so the final count (n = 4.071 citations) does not match the number of 3.438 citations available in 'Records identified from the initial search. Could the authors clarify, please?

3.Results: It would be helpful if the authors could clarify how your results address the paper's two main questions. The narrative categories 1) educating participants, 2) developing learning materials, 3) promoting mentorship, and 4) identifying trainees appear to address the first question. However, I could not find the results for question two [(2) What are the Critical components of How crowdsourcing can be used to improve public health training?] described in detail. It would be relevant if the authors provided a more thorough explanation in this regard.

7. PLOS authors have the option to publish the peer review history of their article (what does this mean?). If published, this will include your full peer review and any attached files.

**Do you want your identity to be public for this peer review?** For information about this choice, including consent withdrawal, please see our Privacy Policy.

Reviewer #1: No

---

## [Editor Report · Decision Letter 2]

30 Jun 2023

Crowdsourcing to support training for public health: A scoping review

PGPH-D-22-01976R2

Dear Professor Tucker,

We are pleased to inform you that your manuscript 'Crowdsourcing to support training for public health: A scoping review' has been provisionally accepted for publication in PLOS Global Public Health.

Best regards,

Diego S Silva, PhD

Academic Editor